# Meat Irradiation: A Comprehensive Review of Its Impact on Food Quality and Safety

**DOI:** 10.3390/foods12091845

**Published:** 2023-04-29

**Authors:** Rossi Indiarto, Arif Nanda Irawan, Edy Subroto

**Affiliations:** Department of Food Industrial Technology, Faculty of Agro-Industrial Technology, Universitas Padjadjaran, Sumedang 45363, Indonesia

**Keywords:** food safety, gamma rays, meat irradiation, meat quality, pathogenic bacteria, shelf life extension

## Abstract

Food irradiation is a proven method commonly used for enhancing the safety and quality of meat. This technology effectively reduces the growth of microorganisms such as viruses, bacteria, and parasites. It also increases the lifespan and quality of products by delaying spoilage and reducing the growth of microorganisms. Irradiation does not affect the sensory characteristics of meats, including color, taste, and texture, as long as the appropriate dose is used. However, its influence on the chemical and nutritional aspects of meat is complex as it can alter amino acids, fatty acids, and vitamins as well as generate free radicals that cause lipid oxidation. Various factors, including irradiation dose, meat type, and storage conditions, influence the impact of these changes. Irradiation can also affect the physical properties of meat, such as tenderness, texture, and water-holding capacity, which is dose-dependent. While low irradiation doses potentially improve tenderness and texture, high doses negatively affect these properties by causing protein denaturation. This research also explores the regulatory and public perception aspects of food irradiation. Although irradiation is authorized and controlled in many countries, its application is controversial and raises concerns among consumers. Food irradiation is reliable for improving meat quality and safety but its implication on the chemical, physical, and nutritional properties of products must be considered when determining the appropriate dosage and usage. Therefore, more research is needed to better comprehend the long-term implications of irradiation on meat and address consumer concerns.

## 1. Introduction

Meat is a valuable element of the human diet as it contains essential elements such as protein, vitamins, and minerals [1,2]. However, these foods are also vulnerable to microbial pathogens and spoilage, posing significant risks to human health [3,4,5]. Ionizing radiation is used in food irradiation to maintain the safety and quality of food items, specifically meat [6].

For decades, food irradiation has been used to reduce microbial contamination and extend the storage period. The procedure entails subjecting the food item to a regulated amount of ionizing radiation, usually accomplished by applying gamma rays, electron beams, or X-rays. [7,8]. The radiation disrupts the DNA and other cellular components of microbes, making them unable to reproduce and causing their death [9,10]. The procedure also breaks down some of the molecules in the food product, which can affect its nutritional quality and sensory properties [11,12,13].

Despite its potential benefits, food irradiation remains controversial, with concerns about its safety, efficacy, and impact on the nutritional quality and sensory properties of food products [14]. Some critics argued that food irradiation could create harmful compounds or destroy essential nutrients [15,16]. In contrast, others questioned the need for irradiation, considering other food safety measures, such as good manufacturing practices and food testing [17,18]. Consumer acceptance of irradiated food products also needs to be addressed, with some people expressing concerns about their safety and acceptability [19,20].

This comprehensive research aims to critically evaluate the existing literature on food irradiation and its repercussions on the quality and safety of meat. The proof of irradiation effectiveness at lowering microbial contamination and prolonging the shelf life of meats is explored along with its potential impact on the physical and chemical characteristics, nutrient content, and sensory properties. This paper will also address the regulatory framework for food irradiation, including labeling requirements and government oversight, as well as identify areas for further research and policy development.

## 2. Sources and Principles of Food Irradiation

Ionizing radiation, such as gamma rays, X-rays, or high-energy electrons, is used to irradiate food [21]. The ionizing radiation wavelength is shown in Figure 1a. Food irradiation is generally determined by the absorbed dose expressed in Gray (Gy) or kilo Gray (kGy), with 1 Gray being equivalent to 1 J/kg of product. The technique is considered a safe and effective way to decrease or eliminate hazardous microbes, prolong shelf life, as well as enhance the quality and safety of food products [22]. The principles of food irradiation are determined by the ability to disrupt the genetic material of microorganisms, preventing them from reproducing or causing illness [23,24]. Figure 1b shows a diagram of how irradiation affects microorganisms’ genetic material (DNA or RNA) directly and indirectly. Direct irradiation can break the bonds between base pairs in the genetic material, killing the cell’s reproduction ability. Then, on the other hand, damage to water molecules creates free radicals and reactive oxygen species, which damage genetic material indirectly [8]. Irradiation also helps to break down certain enzymes and proteins in food that can contribute to spoilage, thereby increasing shelf life [8].

The US, Canada, as well as several European and Asian nations, allow food irradiation [25] using Cobalt-60, cesium-137, and electron-beam accelerators [8,26,27]. Cobalt-60, the most prevalent source of ionizing radiation for food irradiation, is a radioactive isotope that emits gamma rays capable of penetrating deep into food products to destroy harmful microorganisms. Cesium-137 is another source of ionizing radiation, although it is less commonly used than cobalt-60 is [28]. In addition, electron-beam accelerators are used for food irradiation. These devices generate high-energy electrons that can penetrate food products to eliminate harmful microorganisms and extend shelf life [8,29].

Irradiating foods has several benefits, including multifunctional applications as well as guaranteed safety and security [26]. The spectrum produced is effective against bacterial spores across a broad range of concentrations. Given that processing does not involve heat, it is safe for food, does not significantly reduce nutrient levels, leaves no chemical residues, and is simple to control during use. According to Ehlermann [30], to effectively lengthen the lifespan of irradiated food products, the following principles must be observed: (1) *Radurization* uses low doses of 0.1–1 kGy. This amount inhibits respiration, delays ripening, disinfects pests, and inactivates the *Trichinella* parasite. (2) *Radicidation* is referred to as a moderate dose. This radiation uses a quantity of approximately 1–10 kGy, which has the effect of reducing spoilage and microbial pathogens including *Salmonella* sp. and *Listeria monocytogenes*. This dosage is typically found in frozen foods and its application is identical to that of pasteurization, except irradiation does not rely on thermal energy. (3) *Radapertization* uses extremely high doses which are above or equal to 10 kGy, ranging between 30 and 50 kGy. This dose is typically used in the sterilization process because its effect can kill all microorganisms in foodstuffs up to the level of spores. Generally, food irradiation sources and principles are based on the ability of ionizing radiation to disrupt the genetic material of microorganisms, enzymes, and proteins in food products, culminating in improved safety and quality. The use of irradiation is regulated by national and international authorities to ensure its safety and effectiveness.

## 3. The Effects of Irradiation on Meat

### 3.1. Microbial Safety

Microbial safety is a critical aspect of meat production and consumption, as these products can be a source of various harmful microorganisms that can cause foodborne illness [31]. Meat products are potentially contaminated with various pathogens, such as *Salmonella*, *Escherichia coli*, *Campylobacter*, and *Listeria monocytogenes*, leading to severe illness or death in vulnerable populations [31,32].

Contamination might occur at the production, processing, or distribution stage, including on the farm, during transport, in slaughterhouses or processing facilities, and in retail outlets or at home [33,34,35]. Improper handling and storage of meat products can also increase the risk of contamination [36]. Foodborne illness outbreaks related to meat have been reported globally, with various types of products being implicated, including ground beef, chicken, pork, and processed meats [31]. These outbreaks have led to significant public health and economic consequences, highlighting the importance of effective interventions to reduce the risk of contamination.

Irradiation has been studied extensively for its efficacy in reducing microbial contamination of meat [37]. By exposing the food to ionizing radiation, the latter reduces or eliminates harmful microorganisms that can cause foodborne illness [8,37,38]. Previous research showed that irradiation could effectively reduce levels of pathogens such as *Salmonella* and *Escherichia coli* as well as levels of spoilage organisms, leading to improved microbial safety and a reduced risk of foodborne illness [38,39,40,41]. The effectiveness of various types of ionizing radiation on meat, including gamma rays and e-beams, has been studied [42]. According to Park et al. [42], gamma ray irradiation is more effective than e-beam irradiation is at inhibiting microbial growth in meat. Yeh et al. [43] also found that UV light effectively eliminates *Salmonella* spp., *Pseudomonas*, *Micrococcus*, and *Staphylococcus* on meat. The shelf life of meat products is extended by eliminating these contaminant bacteria. Gamma irradiation at low doses can improve microbiological safety, ensure safety, and extend chicken meat’s shelf life without affecting quality [44]. Sedeh et al. [45] found that 3 kGy gamma-irradiated bovine meat reduced the growth of mesophilic bacteria, coliforms, and *Staphylococcus aureus*. The Food and Drug Administration (FDA) determined that a 3.5 kGy gamma ray irradiation dose effectively eliminates pathogenic microbes from fresh meat [46]. Irradiation had the effect of slowing the growth of bacterial cells and deactivating their metabolism [26]. Bacteria are inherently resistant to the effects of irradiation and, in the lag phase or inactive state, will be more resistant. In contrast, those in the growth phase will be more vulnerable [47].

It is essential to note that irradiation is just one aspect of a comprehensive approach to ensuring the microbial safety of meat. Other interventions, such as good manufacturing practices, hygiene controls, and appropriate storage and handling practices, are also crucial for reducing the risk of contamination [48]. In summary, the microbial safety of meat is a critical concern for public health. The risk of contamination can be reduced through various interventions, including using irradiation to mitigate or eliminate harmful microorganisms. However, a comprehensive approach to ensuring food safety, including good manufacturing practices, hygiene controls, and appropriate storage and handling practices, is essential to minimize the risk of foodborne illness. The results of research that investigated how irradiation affected the microbiological qualities of meat are outlined in Table 1. According to these findings, the ability of irradiation to inhibit microbial growth varies depending on the type of meat [49]. The chemical characteristics of meats are affected by the irradiation dose, type and composition of the meat, temperature, gas composition, and microbial load during packaging [50]. When administering irradiation, these factors must be considered.

### 3.2. Chemical Properties

The chemical properties of irradiated meat refer to the changes that occur to the chemical constituents and compositions of the food due to exposure to ionizing radiation [75]. Irradiation can cause both desirable and undesirable effects on the chemical characteristics of meat, depending on the dose and the specific compounds in the food [76].

One of the most significant changes often observed in irradiated meat products is the formation of free radicals. They become reactive molecules that damage cellular components and cause oxidative stress [38]. This leads to lipid oxidation, which causes off-flavors and odors, as well as a decline in nutritional quality due to the loss of essential fatty acids and other nutrients [77,78,79]. However, irradiation at lower doses also aids lipid oxidation by reducing the levels of peroxides and other reactive species [80]. This procedure also affects the protein content of meat, leading to alterations in the composition of amino acids, protein structure, and digestibility [81,82]. These changes have potentially positive and negative effects, mostly on the nutritional value of the food, that are contingent upon the particular proteins involved and the dose of radiation used. The positive effects of irradiation include the fact that irradiation can cause the formation of reactive species, such as free radicals, which can cause the formation of covalent bonds between amino acids in protein molecules. This cross-linking can change the structure of a protein molecule and make it resistant to enzymatic digestion, which causes a decrease in protein digestibility. Irradiation can also cause the denaturation of protein molecules. Denaturation involves opening the protein structure, which can facilitate interactions between amino acids and increase the accessibility of digestive enzymes to protein molecules, and it can also improve protein digestibility [83,84]. However, irradiation can also cause adverse effects; namely, excessive irradiation can cause a breakdown of or changes in amino acid compounds in protein molecules, which causes a decrease in the overall amino acid content and, consequently, decreases protein digestibility [85]. According to Jo et al. [81], electron-beam irradiation at less than 3 kGy did not affect changes in the quality of smoked duck flesh (amino acids, fatty acids, and volatiles) during storage.

Aside from these chemical changes, irradiation also affects the vitamin content of meat products, with some vitamins being more sensitive than others [86,87]. For example, irradiation leads to a loss of vitamin C, while other vitamins, such as vitamin A and E, are relatively stable [87,88]. Irradiation has been shown to alter meat’s oxidation–reduction ability, accelerating lipid oxidation, protein breakdown, and flavor and odor changes [79,89]. When combined with certain antioxidants, such as flavonoids [90,91], irradiation can help prolong the induction period of lipid oxidation. According to Feng et al. [56], storing irradiated meat at 5–10 °C for one week almost did not change the pH, texture, total volatile base nitrogen (TVBN), or microbe number. Meanwhile, Hasan et al. [92] stated that a higher dose of UV irradiation increased 2-thiobarbituric acid (TBA) content, decreased water-holding capacity (WHC), and decreased beef color intensity and tenderness. Hwang et al. [93] found that 2.5 and 5 kGy gamma irradiation reduced nitrite content in chicken sausages and prevented oxidation when combined with antioxidants. The titratable acidity and acid value in meat samples can be reduced by irradiation [51].

Generally, the chemical properties of irradiated meat are complex and depend on various factors, including the dose of radiation, the specific compounds in the meat itself, and processing and storage conditions after the procedure. Future research must completely consider the chemical changes in irradiated meat and how they affect food safety, nutritional quality, and sensory properties. Previous research on how irradiation affects the chemical parameters of meats are summarized in Table 2.

### 3.3. Physical Properties

The physical properties of irradiated meat refer to the changes that occur due to exposure to ionizing radiation. The most common material changes are associated with color, texture, as well as water-holding capacity (WHC) [102].

The color of irradiated meat determines its freshness and consumer acceptance [103]. Color constitutes one of the most noticeable changes commonly observed in irradiated meats [104]. In general, the color of products tends to become lighter or paler, which is attributed to the breakdown of myoglobin and the formation of metmyoglobin, a brownish pigment [104,105,106]. This color change is more pronounced in red than in white meat products, such as beef and poultry, respectively. Several factors affect the color of irradiated meat, including heme pigment concentration (especially myoglobin), oxidation status, ligand formation, and physical characteristics (irradiation dose, pH, temperature, and storage time) [104]. Brugnini et al. [71] discovered that UV irradiation decreased a* values in beef but did not affect L* and b* values. Moreover, Chouliara et al. [97] also found a slight increase in a* values after irradiating chicken breast meat with 4 kGy gamma rays. However, L* and b* values did not change. Irradiation generally increased the a* (redness) value of chicken thighs [107]. Nam and Ahn [108] suggested that carbon monoxide (CO) production during irradiation caused a color change to red. Ramamoorthi et al. [109] found that CO-modified atmosphere-packaged beef irradiated at less than 1.0 kGy appeared redder and has a higher a* value than did beef irradiated at a higher dose. However, on the other hand, Feng et al. [56] found that when raw ground beef was e-beam-irradiated, the a* value decreased as the dose increased. A meat’s pigment concentration decreases due to myoglobin degradation or denaturation as the irradiation dose increases [110].

Texture is another important physical property of meat products affected by irradiation. The procedure causes connective tissue breakdown in meat, leading to a softer texture [78,111]. However, irradiation also leads to a toughening of meat products due to the denaturation of proteins [111,112]. According to Zhao et al. [62], spicy beef jerky samples’ hardness, elasticity, and gumminess decreased with increased gamma ray irradiation. In contrast, Lv et al. [100] stated that irradiation at 3–5 kGy using an e-beam improved the texture of *Tegillarca granosa* meat. Rodrigues et al. [113] found that combining 9 kGy gamma irradiation and aging could improve tenderness in *Nellore beef.* Increasing the irradiation dose causes a decrease in the myofibrillar protein content in meat. It changes the secondary structure, which causes changes in the functional properties of the myofibrillar protein, affecting the texture of the meat [101]. The dose, temperature, pH, packaging method, storage time, muscle type, and water content all influence these changes [114]. Furthermore, the sensitivity of different types of proteins to the effects of irradiation on meat texture varies [101].

Water-holding capacity (WHC) refers to the capability of food to water retention during cooking or processing [115]. Irradiation can affect the water-holding capacity of meat products by causing protein structure changes [38]. This might lead to the loss of water during cooking or processing, affecting the quality of the final product [116]. Hasan et al. [92] found that beef samples irradiated with UV at higher doses showed lower water loss during cooking and lower WHC percentages.

Generally, the physical properties of irradiated meats are essential considerations in ensuring their quality and consumer acceptance. Understanding the changes in the physical properties due to irradiation is necessary for developing effective processing methods and enhancing quality. Table 3 presents various research studies which examined the impact of irradiation on the physical attributes of meat.

### 3.4. Shelf Life Extension

Shelf life extension is a critical aspect of meat product preservation, as it is essential for maintaining the quality and safety of these perishable items over a desired storage period [117,118]. Meat spoilage and degradation occur due to various factors, including pathogenic microorganisms, enzymatic activity, as well as oxidation [119,120,121]. These factors alter the color, texture, flavor, nutrient content, and growth of harmful microorganisms that can cause foodborne illness [122,123,124].

Irradiation is one method that has been investigated extensively for its effectiveness at extending the shelf life of meat. Ionizing radiation when applied to food minimizes the levels of or completely eradicates pathogenic microbes that cause spoilage and degradation [8]. This procedure has been reported to also lengthen shelf life and enhance product safety [25].

Previous research showed that irradiation was a good way to make meats such as ground beef, chicken, and pork last longer [54,70,75,125,126,127]. This method reduces or eliminates pathogenic microbes including *Salmonella* and *Escherichia coli*, as well as spoilage organisms such as *Pseudomonas* and lactic acid bacteria [41,54,128]. This can significantly extend the shelf life of food products. Compared to other methods of meat preservation, such as thermal processing and chemical preservatives, irradiation is a safe and effective technique [129]. Furthermore, irradiation has been reported to slightly impact the sensory attributes and nutritional value of a product, making it a desirable preservation method [11,130].

According to a study by Damdam et al. [54], the shelf life of beef, chicken, and salmon fillets was extended by 66.6% through a combination of vacuum packaging and UV irradiation. The study conducted by Gabelko et al. [131] investigated the impact of e-beam irradiation at 4 kGy on the shelf life of chicken meat while also considering its effect on sensory characteristics. The results indicated that the treatment extended the shelf life of the meat without any significant impact on its sensory attributes. According to Sedeh et al. [45], the shelf life of irradiated bovine meat stored at 4–7 °C and exposed to 3 kGy irradiation was 14 days. In contrast, non-irradiated bovine meat had only three days of shelf life. According to Derakshan et al. [75], the recommended doses of electron-beam irradiation for maintaining quail meat’s quality and shelf life are 1.5 and 3 kGy.

There are limitations to the effectiveness of irradiation; for example, the process is less effective at controlling certain types of spoilage bacteria that do not have high levels of DNA damage. Although irradiation does not introduce harmful chemicals or residues into the food product, there are still concerns regarding its potential impact on food quality and safety [19,25]. Irradiation in general is promising for meat shelf life extension but it is only one aspect of a comprehensive preservation strategy that includes proper handling, storage, and other food safety interventions. Previous research has been conducted to determine how irradiation impacts the storage life of meat and their findings are summarized in Table 4.

### 3.5. Nutritional Quality

The nutritional quality of meat is essential for consumers, while irradiation as a processing method raises concerns about the potential changes to nutrient composition and bioavailability. Previous research showed that irradiation could cause some changes in the nutrient composition of meat [134]. For example, the thiamine content of pork and beef has been shown to decrease after irradiation, while the levels of other B vitamins have been shown to remain stable [87,135,136,137]. The effects of irradiation on mineral content could be more precisely elucidated, with some research reporting a decrease in certain minerals while others reported no significant changes [138,139]. Some research also reported that irradiation caused changes in protein quality, leading to reduced digestibility and bioavailability [85,140], while others found no significant effects [141]. Excessive irradiation can cause the oxidation of proteins, which causes the formation of carbonyl groups, breaking of double bonds, or can even change the peptide bonds that exist in protein structures. These changes will cause the protease enzymes that play a role in protein digestion to be unable to recognize and digest proteins so that the digestibility and bioavailability of proteins decrease [142]. No significant effect occurs if irradiation is only given at low intensity and for a short time so that the energy from the irradiation light is not enough to change the structural configuration of the protein [141]. According to Zhao et al. [141] irradiating spicy yak jerky with electron-beam irradiation at a maximum dose of 7 kGy did not significantly change the protein quality measured by total amino acid content (TAA). However, a dose of 9 kGy showed a significant decrease in TAA. Stadtman and Levine [143] investigated how free radicals generated by irradiation affected free amino acids and their residues. They found that free radicals caused amino acids such as lysine, arginine, proline, cysteine, threonine, leucine, and histidine to change to carbonyl derivatives and other chemicals. Higher irradiation doses make more free radicals and cause a loss of more amino acids [141].

Lipid oxidation is another potential concern with irradiated meat products, as it can lead to changes in flavor, texture, and nutritional quality [144]. Reports showed that irradiation increased lipid oxidation in meat [77,144]. These changes may vary depending on the type of meat, radiation dose, and storage conditions [77]. Jia et al. [145] found that different irradiation doses can change the meat system’s redox potential, accelerating fat and protein oxidation and changing the meat’s color, taste, and flavor. According to Huang et al. [146] smoked chicken breast gamma-irradiated at >3 kGy had reduced protein oxidation and increased lipid oxidation. Zhang et al. [147] also found that marbled beef lipids were significantly affected by irradiation, but increasing the dose from 2.5 to 4.5 kGy had little effect. Lipids in meat consist of saturated and unsaturated fats. Unsaturated fats (oleic, linoleate, linolenic, and arachidonic) are easily oxidized if subjected to prolonged heating or irradiation. The high irradiation energy will cause the release of electrons in the double bonds of unsaturated fat molecules to produce peroxyl radicals. Peroxyl radicals can then undergo propagation or multiplication, also known as the autoxidation process. In further oxidation processes, peroxyl radicals can produce volatile secondary products, causing rancid flavors or off-flavors to form in meat, thereby reducing meat quality [77,148].

Irradiation also affects the stability of vitamins in meat [87], with some research reporting that it causes a decrease in the levels of some vitamins, such as vitamin B complex and vitamin C, while those of others, including vitamin E, remain stable [87,149]. Vitamins degrade quickly due to temperature, light, oxygen, water alkalinity, pH, and contact with other components [150]. Meat contains water-soluble B-complex vitamins such thiamin (B1), riboflavin (B2), niacin (B5), pyridoxine (B6), biotin (B10), cobalamin (B12), choline, folic acid, and pantothenic acid. Fat-soluble vitamins are more stable under irradiation than water-soluble vitamins are [87]. There are various possible mechanisms of irradiation for decreasing vitamin levels in meat. The breakdown of vitamins caused by irradiation reduces the vitamin content in meat, reducing its nutritional value. Reactive oxygen species (ROS) are free radicals produced by irradiation [38] that can react with vitamins to cause damage or oxidation. Furthermore, radiolytic products can react with vitamins, causing damage [151]. They can also react with protein or lipid components, altering their nutritional and sensory properties [152]. However, vitamin irradiation generally depends on the dose of irradiation used, the type of vitamin, and the presence of other food components [87]. Gabelko and Sapozhnikov [131] exposed poultry and vegetables to 4 kGy e-beam irradiation. They discovered that irradiation significantly affected vitamin C levels, decreasing it, but there was no significant difference after one month of storage. Additionally, Dionisio et al. [87] reported that after 6.0 kGy irradiation, the vitamin B1 content of meat products was reduced by 47%, whereas riboflavin and niacin acid (nicotinic acid) content was more radiation resistant.

Despite these potential changes in nutrient composition and bioavailability, the safety of irradiated meat foods for human consumption has been extensively studied and confirmed. The World Health Organization (WHO), the US Food and Drug Administration (FDA), as well as other regulatory bodies concluded that irradiated meat products were safe for human consumption and had no adverse effects on health [19,153].

Finally, while irradiation might potentially cause several changes in the nutrient composition and quality of meat, it is a safe and effective method for reducing the risk of foodborne illness. More research is required to better understand the consequences for nutrient composition and bioavailability as well as to optimize the processing parameters to minimize any potential adverse effects on nutritional quality. The results of several investigations on how irradiation affects the nutritional profile of meat are summarized in Table 5.

### 3.6. Sensory Properties

The sensory properties of meats are crucial in determining their overall quality and consumer acceptance [155]. Food irradiation, a preservation technique that utilizes ionizing radiation to reduce microbial contamination and extend shelf life, has contributed to the increased sensory attributes of irradiated meat [8]. These changes include color, texture, and flavor alterations, which can affect consumer acceptance and perceptions of the products [19].

Irradiation up to 10 kGy does not affect nutritional properties or food safety [75]. Irradiation accelerates lipid oxidation and discoloration, and decreases the levels of sensory properties that cause off-flavors in meat and meat products [156]. According to research by Huang et al. [146], gamma ray irradiation of smoked chicken breast at 2 kGy had no discernible impact on sensory attributes. However, irradiation at 3, 4, and 6 kGy caused a decline in sensory attributes. Irradiation dose is positively correlated with TBARS content and drip loss is closely correlated with texture, affecting sensory scores. It was discovered that TBARS content and irradiation had a positive correlation. A higher level of lipid oxidation impacts flavor and taste because more flavor and taste compounds are produced. Furthermore, the degree of drip loss is related to meat texture and influences texture scores [146].

Several factors, including knowledge about the irradiation process, the perceived safety of it, and the sensory properties of the products, influence consumer acceptance of irradiated meat. According to previous research, consumers will buy irradiated meat products if they are labeled correctly, appear safe, and are of a high quality [157]. However, some consumers may have negative perceptions, presumably due to the products’ sensory properties [158].

Different strategies have been proposed to keep improving the sensory acceptability of irradiated meat, including the use of packaging and storage conditions that can minimize the sensory changes caused by irradiation [159], the use of flavor enhancers and other additives to enhance the flavor and aroma of the products, as well as the selection of meat cuts and processing techniques that are less reliant on irradiation [160].

The sensory properties of irradiated meat are an essential consideration in developing and marketing these products; hence, strategies for improving meats’ sensory quality should be carefully considered. The results of previous research on the impact of irradiation on the sensory qualities of meat are presented in Table 6.

## 4. Regulatory Framework for Food Irradiation

Food irradiation is governed by national and international regulations that aim to ensure the safety and efficacy of the technology [25]. In the US, the FDA and USDA regulate irradiated food safety and labeling [162]. These foods must be labeled “treated with radiation” or something similar, and the dose must be mentioned [163,164,165].

The Codex Alimentarius Commission, a joint body of the FAO and WHO, has established food irradiation guidelines and standards internationally. These guidelines guide food irradiation for specific products, including meat and meat, and establish maximum permitted irradiation doses.

Aside from regulatory oversight, government agencies ensure the safety and quality of irradiated meat products [162]. The USDA Food Safety and Inspection Service (FSIS) ensures the safety and labeling of meat as well as poultry products, including irradiated ones. The agency inspects and tests irradiated products to ensure they meet regulatory requirements and are safe to eat.

According to General Standard, the minimum absorbed dose must be sufficient to achieve the technological objective. The maximum dose must be less than that which would endanger consumer safety or adversely affect structural, functional, nutritional, or sensory characteristics [166]. The Codex Alimentarius Commission has declared food irradiated with up to 10 kGy gamma rays safe, so toxicological testing is no longer needed. The United Nations affirms that food can be treated at any dose without causing harm. When carried out correctly, high-dose irradiation can be used on various foods to improve their hygienic quality, maintain stability, and create special products [25]. Radiological safety, toxicological safety, microbiological safety, and nutritional adequacy are the FDA’s criteria for irradiated food safety. Food irradiation-related radioactivity and toxicity were not found to exist. Radiation has been approved to eliminate the most resistant bacteria, *Clostridium botulinum*, and its toxin without increasing microbiological risks [167].

The regulatory framework for food irradiation ensures the safety and efficacy of the technology, as well as provides consumers with clear and accurate product information. Government agencies must enforce these regulations to ensure the safety and quality of irradiated meat.

## 5. Research Needs and Future Directions

When discussing the future directions and research requirements for the issue of food irradiation on meat, a comprehensive review usually centers on potential advancements and unresolved areas of uncertainty within the domain. This section aims to provide insight into the research needs to understand the impact of irradiation on meat products and identify areas of future investigations.

Emerging technologies and applications for food irradiation are a significant focus in this section. New techniques of food irradiation that may offer potential benefits over traditional methods are being explored, including pulsed electron beams, X-rays, and gamma radiation. Additionally, the potential of using irradiation to prolong the shelf life of pre-packaged meat products is an area of active investigation.

Combining irradiation with other processes can improve food safety, nutritional value, product quality, and losses during commercialization [168,169,170]. Irradiation with plant essential oils, modified atmosphere packaging, or mild heat treatment increased relative bacterial radiosensitivity (RBR) by 2–4 times [170,171,172]. This combination reduces the radiation dose, heat, nutrition, and sensory qualities compared to those of natural products, resulting in a better product. Irradiation technology combined with mild heat treatment can reduce or eliminate bacteria and parasites to extend the shelf life of nutritionally sound products [173]. Furthermore, combining nano-bacteriostatic agents with microwave technology, UV irradiation, ultrasonic treatment, and other processing technologies is widely carried out for agricultural and aquatic products. Nano-bacteriostatic agents are being used as food supplements to improve meat sterilization. Xu et al. [174] found that combining these technologies for processing can reduce thermal processing intensity and protect a product’s nutritional and organoleptic quality. More research is required to develop better processing methods and preserve meat products.

Areas of uncertainty and controversy in food irradiation may also be addressed. For example, according to some research, irradiation can lead to the formation of potentially harmful substances in meat products, while others suggest that this risk is minimal. Future research and experimentation should seek to clarify these issues.

Finally, the section on future directions and research needs may identify specific areas where more investigations are required to better understand the influence of ionizing radiation on meat. This might include research investigating the impact of irradiation on sensory quality and examining the nutritional composition of irradiated meat products. Additionally, there is a need to explore the potential repercussions of irradiation on specific types of meat products, such as ground meat or poultry.

## 6. Conclusions

Food irradiation is a promising technology that has the potential to improve both the quality and the safety of meat. Current research suggests that irradiation can reduce microbial contamination, extend shelf life, and maintain the nutritional quality of meat products. However, the sensory properties may be negatively affected, and there is a need for further research to address this issue. It is also important to note that national and international agencies regulate irradiation in food processing, and there are labeling requirements for irradiated meat products. Government agencies have an essential function in ensuring the safety and quality of consumers. Future research on food irradiation could also prioritize finding emerging technologies and applications, addressing areas of uncertainty and controversy, as well as improving current knowledge about the effects of irradiation on meat. Considering the current evidence, it is recommended that policymakers consider incorporating irradiation as part of a comprehensive food safety strategy. Additionally, stakeholders in the meat industry should consider investing in research and innovation to enhance the organoleptic characteristics of products and increase consumer acceptance.

## Figures and Tables

**Figure 1 foods-12-01845-f001:**
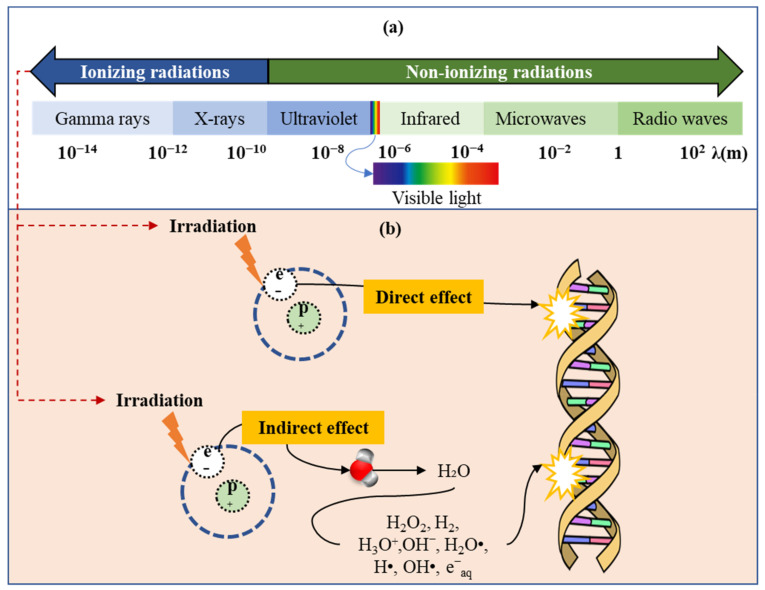
(**a**) Electromagnetic spectrum; (**b**) schematic of the effect of irradiation on nucleic acids.

**Table 1 foods-12-01845-t001:** Impact of irradiation on the microbiological characteristics of meat.

Sample	Treatment	Dose	Sample Storage Conditions	Microbiological Effect(s)	Ref.
Ground beef	UV irradiation + bacteriophage	800 µW/cm^2^	Refrigerator storage at 4 °C	The most effective approach to reduce *Salmonella* in ground beef samples was a combination of UV radiation and bacteriophage, resulting in a 99% reduction (2 log cycles) in levels of the pathogen.	[43]
Smoked guinea fowl meat	Gamma irradiation	2.5, 5 and 7.5 kGy	Stored in the refrigerator (2–4 °C); storage period of 7 days	The amount of *Staphylococcus* sp., *Escherichia coli*, and *Bacillus* sp. decreased as the irradiation dose and storage period increased.	[51]
Bovine meat	Gamma irradiation	0, 0.5, 1, 2 and 3 kGy	Commercial freezing bags; storage: 3 weeks in refrigerator (4–7 °C), and 8 months in freezer (−18 °C)	Mesophilic bacteria*,* coliforms, and *Staphylococcus aureus* content was reduced with 3 kGy irradiation.	[45]
Sliced ham	UV-C irradiation	408, 2040, 4080, and 6120 mJ/cm^2^	Refrigerated storage; 0, 7, and 14 days	Irradiation effectively reduced the growth of *Yersinia enterocolitica* and *Brochothrix thermosphacta*, as well as total microbial count, without any significant impact on the quality of the product.	[52]
Beef sausage patties	Gamma irradiation and electron-beam irradiation	0–20 kGy	Vacuum-packed, stored at 30 °C; 10 days	After 5 kGy and 15 kGy irradiation, the bacterial count was less than 2 log CFU/g.	[42]
Broiler chicken	Gamma irradiation	0–5 kGy	Packed in plastic + sealer, cold storage	*Escherichia coli* and *Staphylococcus aureus* were inhibited by 5 kGy irradiation.	[53]
Beef, Chicken and Salmon Fillets	UV-C irradiation	360 J/m^2^	Vacuum storage; 5 days	Vacuum packing combined with UV-C irradiation was very effective for slowing the growth of *Pseudomonas* sp., aerobic bacteria, lactic acid bacteria*, Salmonella*, and *Escherichia coli* during storage.	[54]
Chicken breast	UV-C irradiation	0.12–3.6 J/cm^2^; 0–15 min	Placed on the tray; storage in the refrigerator; 30 min	The exposure of chicken breast to irradiation for a minimum of 1 min led to a reduction of over 90% in *Salmonella*.	[55]
Raw ground beef	Electron-beam irradiation	0, 1.5, 3.0 and 4.5 kGy	Vacuum packed for 24 h at 4 °C	Dimethyl sulfoxide, a characteristic off-odor compound in irradiated meat, was detected in samples produced from the radiolysis of methionine when subjected to 4.5 kGy irradiation.	[56]
Ground beef	Gamma irradiation	First assay: 0.26, 0.44, 0.67 and 0.86 Second assay: 1 kGy Third assay: 2.5 kGy	Freezer storage at −18 °C	1 kGy reduced Shiga toxin-producing *Escherichia coli* (STEC) by 5 log CFU/g in ground beef.	[57]
Minced camel meat	Gamma irradiation	0, 2, 4, and 6 kGy	Refrigerator storage at 1–4 °C for 2, 4, and 6 weeks	Irradiation increased shelf life from 2 weeks to 6 weeks by reducing mesophilic aerobic plates and coliforms.	[58]
Raw ground beef patties	Electron beam irradiation	2, 4, and 6 kGy	Packed with nylon polyethylene bag; stored at −18 °C; storage period: up to 7 months	Exposing the sample to electron beams for a dosage of 2 kGy had reduced the microbial population to a safe level.	[59]
Beef trimmings (20% fat)	Gamma irradiation	2 to 5 kGy	Sterile bags; polyethylene bags; stored at −18 °C or 2 °C; up to 30 days	Pathogenic microorganisms like *Listeria monocytogenes* and *Escherichia coli* can be reduced by 2.5 kGy irradiation.	[60]
Broiler meat	UV irradiation, activated oxygen (AO)	9.4, 18.8 dan 32.9 mW/s per cm^2^	Up to 12 days in vacuum bags at 3–5 °C	UV irradiation alone or in combination with AO affected *Campylobacter jejuni* survival. *Campylobacter jejuni* was reduced most when irradiated at 32.9 mW/s per cm^2^.	[61]
Spicy beef jerky	Gamma irradiation	0, 0.5, 1.5, 3, 4, 6, and 8 kGy	Vacuum packed; refrigerator storage (4–8 °C); 1 week	Irradiation at a minimum of 1.5 kGy resulted in *Escherichia coli* levels lower than 30 MPN/100g. A minimum irradiation dose of 6 kGy led to aerobic bacteria levels lower than 10 CFU/g.	[62]
Chicken breast fillets	UV-C irradiation; caffeine application (20 mM/g)	0–15 J/cm^2^	Styrofoam boxes; stored at −10 °C	The inactivation of *Escherichia coli* was found to increase with higher concentrations of caffeine and UV-C doses.	[63]
Chicken breast fillets	UV-C irradiation; caffeine application (20 mM/g)	0–15 J/cm^2^	Styrofoam boxes; stored at −10 °C	The combination of caffeine + UV-C can reduce *Escherichia coli* by more than 6 logs.	[63]
Meat (beef and chicken)	UV-C and curcumin-mediated photodynamic inactivation (PDI)	0.8, 1.6, 2.3, 3.1, 3.9 and 7.8 J/cm², for 1, 2, 3, 4, 5 and 10 min	Placed on a stainless steel grid	Meats treated with UV-C radiation and PDI had significantly reduced levels of *Escherichia coli* and *Staphylococcus aureus*.	[64]
Chicken meat	UV-C irradiation	0.62, 1.13, and 1.95 mW/cm²	Packaged in polyvinyl chloride film and kept at 4 °C for 9 days	Irradiation reduced pathogenic bacteria at 1.95 mW/cm^2^ without affecting chicken meat quality. The highest UV-C intensity reduced the initial bacterial load and extended the lag phase and shelf life of meat products.	[65]
Blue swimming crab meat (*Portunas pelagicus*)	Gamma irradiation	2, 4, and 6 kGy	Stored at 0–4 °C in polypropylene bags for 28 days	*Listeria monocytogenes* were inactivated by 4 and 6 kGy gamma irradiation.	[66]
Lamb loin cuts (*Longissimus dorsi*)	Gamma irradiation	1.5 and 3.0 kGy	Vacuum packed with EVA/PVDC plastic bags; stored at 1 ± 1 °C for 56 days.	Microorganisms were reduced in a 3.0 kGy extended lamb loin stored at 1 °C for 14 to 56 days without affecting physicochemical properties or consumer acceptance.	[67]
Turkey breast meat	Gamma irradiation	0.0, 0.5, 2.0 and 4.0 kGy	2 months at −18 °C	*Mesophilic* and *coliform* bacteria were reduced by over 5 log units with a 4 kGy dose. The shelf life was six months because salmonella was not present.	[68]
Rabbit meat	Gamma irradiation	0, 1.5, and 3 kGy	Packed in polyethylene pouches; stored in the refrigerator 3–5 °C; 3 weeks	In meat samples, 3 kGy irradiation inhibited *Staphylococcus aureus*, *Listeria monocytogenes*, *Salmonella*, and *Enterobacteriaceae* growth. It extended meat shelf life without affecting sensory quality.	[69]
Raw minced beef meat	Gamma irradiation	2, 4, 6, 8, and 10 kGy	Sterile bags; stored in refrigerator 4–5 °C for 28 days	The growth of *Staphylococcus aureus*, *Salmonella* sp., *Shigella* sp., and other dangerous microorganisms were significantly reduced, while beef shelf life was increased by over four weeks.	[70]
Beef	UV-C irradiation	165, 330 and 398 mJ/cm^2^	Vacuum-packed; stored at 4 °C for 8 weeks	In vacuum-packed meat, lactic acid and irradiation inhibited the growth of *L. monocytogenes* and delayed lactic acid bacteria growth for 2 weeks. This method preserved meat color for up to eight weeks at 4 °C.	[71]
Beef loins	Gamma irradiation; chitosan film containing cumin essential oil nanoemulsion	2.5 kGy	Stored at 3 °C for 21 days in polyethylene pouches.	The combination of these treatments effectively controlled the growth of microbial flora and inoculated foodborne pathogens.	[72]
Goat meat	Gamma irradiation	4 kGy	Packed in polyethylene bags without vacuum; refrigerator storage at 2–4 °C; 8 days	Irradiation reduced total aerobic bacteria*,* psychrotrophic*,* total staphylococci, yeast, and molds. Irradiated samples had no enterococci, *Escherichia coli*, or *Staphylococcus aureus*.	[73]
Meatball	Gamma irradiation	2 and 4 kGy	Aerobic packaging and MAP; refrigerator storage (2–4 °C); 0–14 days	Using 5% O_2_ and 50% CO_2_ modified atmosphere packaging (MAP) and up to 4 kGy irradiation, *Escherichia coli O157:H7*, *Salmonella enteritidis*, and *Listeria monocytogenes* growth was reduced.	[74]
Smoked guinea fowl meat	Gamma irradiation	2.5, 5 and 7.5 kGy	Stored for 7 weeks in the refrigerator (2–4 °C)	*Staphylococcus aureus*, *Serratia marcescens*, and *Enterobacter cloacae* growth decreased with increasing irradiation doses.	[51]

**Table 2 foods-12-01845-t002:** Impact of irradiation on the chemical characteristics of meat.

Sample	Treatment	Dose	Sample Storage Conditions	Chemical Characteristics	Ref.
Smoked guinea fowl meat	Gamma irradiation	2.5, 5 and 7.5 kGy	Stored in the refrigerator (2–4 °C); storage period of 7 days	The titratable acidity and acid values in meat samples were reduced by irradiation.	[51]
Raw ground beef	Electron-beam	0, 1.5, 3.0 and 4.5 kGy	Vacuum packed and kept at 4 °C for 24 h	After one week at 5–10 °C, the meat’s pH, texture, TVBN (total volatile basic nitrogen), and growth of microorganisms did not change. Therefore, the irradiated products had a longer shelf life.	[56]
Minced camel meat	Gamma irradiation	0, 2, 4 and 6 kGy	Polystyrene trays covered in polyethylene film, stored in a refrigerator at 1–4 °C for up to 6 weeks	Camel meat irradiation slightly affected TVBN and lipid oxidation.	[58]
Beef trimmings (20% fat)	Gamma irradiation	2 to 5 kGy	Sterile bags; polyethylene bags; stored at −18 °C or 2 °C; stored for up to 30 days	5 kGy irradiation increased the off-flavor intensity of 30-day-aged trimming patties.	[60]
Beef loins	Gamma irradiation; chitosan film containing cumin essential oil nanoemulsion	2.5 kGy	Stored at 3 °C for 21 days in polyethylene pouches	Irradiation and chitosan-containing essential oil nanoemulsion showed slowed changes in water, protein, total lipid, and ash content.	[72]
Smoked guinea fowl meat	Gamma irradiation	2.5, 5 and 7.5 kGy	Stored in the refrigerator (2–4 °C) with a storage period of 7 weeks	Irradiation decreased the titratable acidity and the acid value of meat samples.	[51]
Minced camel meat	Gamma irradiation	0, 2, 4 and 6 kGy	Placed in polystyrene trays and covered with polyethylene film; stored in refrigerator 1–4 °C; storage for up to 6 weeks	Irradiation did not affect camel meat moisture, protein, fat, thiobarbituric acid (TBA) values, total acidity, or fatty acid content.	[58]
Rabbit meat	Gamma irradiation	0, 1.5, and 3 kGy	Packed in polyethylene pouches; stored in the refrigerator 3–5 °C; stored for 3 weeks	Irradiating rabbit meat samples increased the content of thiobarbituric acid reactive substances (TBARS) but did not affect total volatile nitrogen (TVN). Both irradiated and non-irradiated samples had significantly increased TBARS content and TVN during storage.	[69]
Turkey breast meat	Gamma irradiation	0.0, 0.5, 2.0 and 4.0 kGy	Stored at −18 °C; 2 months	Irradiation increased peroxide values but no significant effect was found on TVN content.	[68]
Chicken breast fillets	UV-C irradiation; caffeine application (20 mM/g)	0–15 J/cm^2^	Styrofoam boxes; stored at −10 °C	UV-C irradiation and caffeine treatment did not affect protein, crude fat, moisture, ash, total acidity, pH, and absorption coefficient.	[63]
Raw ground beef	Electron beam	0, 1.5, 3.0 and 4.5 kGy	Vacuum packed, stored at 4 °C for 24 h	From 0 to 4.5 kGy, irradiation doses increased lipid and protein oxidation by 156% and 64%, respectively. It suggested that the sample was more prone to lipid oxidation than protein oxidation.	[56]
Chicken meat	UV-C light	0.62, 1.13, and 1.95 mW/cm²	Polyvinyl chloride-wrapped; stored at 4 °C for 9 days	UV-C light increased tyramine, cadaverine, and putrescine levels.	[65]
Bovine meat	Gamma irradiation	0, 0.5, 1, 2 and 3 kGy	Packed in commercial freezing bags; refrigerator storage at 4–7 °C for 3 weeks and freezer storage at −18 °C for 8 months	The chemical characteristics of bovine meat samples subjected to irradiation, specifically the total nitrogen and peroxide values, exhibited no significant differences during frozen storage.	[45]
Chicken Meat	Gamma irradiation, kale leaf powder (KLP) and their combination	3 kGy	Aerobically packaged; refrigerator storage at 4 °C; up to 14 days	1% and 2% KLP decreased chicken meat pH during storage, but 3 kGy gamma radiation increased pH. The pH of the meat decreased when KLP was combined with 3 kGy irradiation.	[94]
Korean native cattle (*Hanwoo*) beef	UV irradiation	4.5 mW s/cm^2^	Wrapped in polyvinyl chloride; stored at 3–5 °C; stored for 9 days	During storage, irradiated meat had higher TBA values. Volatile basic nitrogen (VBN) values did not change significantly. The physico-chemical properties of the samples were unaffected by UV radiation.	[95]
Spicy beef jerky	Gamma irradiation	0, 0.5, 1.5, 3, 4, 6, and 8 kGy	Vacuum-packed; refrigerator storage (4–8 °C); 1 week	The content of capsanthin in the spicy beef jerky sample decreased, while the level of TBARS increased with an increase in irradiation dose	[62]
Meatball	Gamma irradiation	2 and 4 kGy	Packaged in aerobic packaging and MAP; refrigerator storage 2–4 °C; 0–14 days	The TBARS value of the meatballs showed a significant increase as the irradiation dose increased.	[74]
Chicken meat	Gamma irradiation, 0.1 % chitosan and their combination	0.5 kGy	Packed in polypropylene bags, chilled conditions 4–6 °C; storage up to 14 days.	After 11 days, treated samples showed significant changes in TBARS, trichloroacetic acid-soluble protein (TSP), and trichloroacetic acid (TCA) content.	[96]
Goat meat	Gamma irradiation	4 kGy	Packed in polyethylene bags without vacuum; refrigerator storage at 2–4 °C; 8 days	Compared to non-irradiated goat meat samples, irradiation decreased pH, water-holding capacity, and TBA. However, irradiated samples had higher free fatty acids and a* values.	[73]
Beef, Chicken and Salmon Fillets	UV-C irradiation	360 J/m^2^	Vacuum stored; 5 days	The pH of all samples remained unchanged after exposure to irradiation.	[54]
Beef	UV-C irradiation	165, 330 and 398 mJ/cm^2^	Vacuum-packed; stored at 4 °C; 8 weeks	Compared to non-irradiated meat, irradiated meat had a lower pH.	[71]
Chicken breast meat	Gamma irradiation	4 kGy	Packaged under MAP; refrigeration storage at 2–6 °C; storage period of 25 days	The TBA value of meats remained below 1 mg of malondialdehyde per kg after 25 days of storage.	[97]
Ground beef patties	E-beam irradiation	1.5, dan 2.0 kGy	Insulated containers; refrigerator storage (4 °C); 0–28 days	In irradiated samples, dried plum puree, rosemary extract, and BHA/BHT antioxidants reduced TBARS levels.	[98]
Meatballs	Gamma irradiation	20, 25 dan 35 kGy	Vacuum-packed + frozen 24 h; room temperature (irradiated samples) and freezer (non-irradiated); 2 months	After irradiation, the pH of meatballs increased before storage but decreased after two months. Irradiated samples had significantly lower Aw values.	[99]
Korean native cattle (*Hanwoo*) beef	UV irradiation	4.5 mW s/cm^2^	Wrapped in polyvinyl chloride; stored at 3–5 °C; 9 days	During storage, pH was not significantly different in all irradiated beef samples.	[95]
Chicken, turkey and mixed ground meat	X-ray irradiation	0.5, 1, 3 and 5 kGy	Plastic bag; stored at −80 °C	Exposure of the meat sample to irradiation led to alterations in the lipidome.	[100]
*Tegillarca granosa* meat	Electron-beam irradiation	0, 1, 3, 5, 7 or 9 kGy	Vacuum-packed	Irradiation at 7 kGy enhanced the biochemical characteristics of proteins in the meat sample.	[101]

**Table 3 foods-12-01845-t003:** Impact of irradiation on the physical characteristics of meat.

Sample	Treatment	Dose	Sample Storage Conditions	Physical Characteristics	Ref.
Spicy beef jerky	Gamma irradiation	0, 0.5, 1.5, 3, 4, 6, and 8 kGy	Vacuum-packed; refrigerator storage (4–8 °C); 1 week	The hardness, elasticity, and gumminess of spicy beef jerky samples decreased with increased irradiation.	[62]
Beef	UV-C irradiation	165, 330 and 398 mJ/cm^2^	Vacuum-packed; stored at 4 °C; 8 Weeks	Irradiation reduced a* values, but L* and b* values were unaffected. After 8 weeks of storage, L* and a* values were lower than those of non-irradiated samples, but b* values were not significantly different.	[71]
Chicken breast meat	Gamma irradiation	4 kGy	Packaged under MAP; refrigeration storage at 2–6 °C; storage period of 25 days	The irradiation treatment led to a slight increase in the a* values, whereas no significant change was observed in the L* and b* values.	[97]
Korean native cattle (*Hanwoo*) beef	UV irradiation	4.5 mW s/cm^2^	Wrapped in polyvinyl chloride; stored at 3–5 °C; 9 days	During storage, there was an increase in L* and b* values, while the a* value in all samples initially increased until day six and then decreased.	[95]
Blue swimming crab meat (*Portunas pelagicus*)	Gamma irradiation	2, 4, and 6 kGy	Packed in polypropylene bags; stored at 0–4 for up to 28 days	A low dose of gamma irradiation enhanced the safety of crab meat without causing any undesired changes in texture and L^⁎^ color value.	[66]
Raw ground beef	Electron-beam irradiation	0, 1.5, 3.0 and 4.5 kGy	Vacuum-packed, stored at 4 °C for 24 h	The color of the samples faded upon irradiation at 4 kGy.	[56]
Broiler meat	UV irradiation; activated oxygen (AO)	9.4, 18.8 and 32.9 mW/s per cm^2^	Vacuum bags; stored at 3–5 °C for up to 12 days	The color of the samples treated with UV alone or combined with AO did not exhibit any significant differences.	[61]
Beef trimmings (20% fat)	Gamma irradiation	2 to 5 kGy	Sterile bags; polyethylene bags; stored at −18 °C or 2 °C for up to 30 days	Up to 5 kGy of irradiation did not affect beef patties’ L*, a*, and b* values.	[60]
Meatball	Gamma irradiation	2 and 4 kGy	Packaged in aerobic packaging and MAP; refrigerator storage at 2–4 °C; 0–14 days	During the first 7 days of storage, no significant changes were observed in the a* value, but after 14 days, there was a considerable decrease. In contrast, irradiation did not affect the L* and b* values.	[74]
Spicy beef jerky	Gamma irradiation	0, 0.5, 1.5, 3, 4, 6, and 8 kGy	Vacuum packed; refrigerator storage (4–8 °C); 1 week	The degree of lightness, amount of drip loss, and presence of an off-odor increased as the irradiation dose was increased.	[62]
Meatballs	Gamma irradiation	20, 25 dan 35 kGy	Vacuum-packed + frozen 24 h; room temperature storage (irradiated samples) and freezer storage (non-irradiated samples); 2 months	Meatball firmness decreased as the irradiation dose increased, but L* color values initially did not. After two months of storage, irradiated samples had significantly higher L* values.	[99]
Korean native cattle (*Hanwoo*) beef	UV	4.5 mWs/cm^2^	Wrapped in polyvinyl chloride; stored at 3–5 °C; 9 days	During storage, no significant differences in drip loss and shear force parameters existed among all irradiated beef samples.	[95]
Fresh beef	UV irradiation	12.7, 25.5 and 38.2 W.s/cm²	Packed in polyethylene pouches, and stored in the refrigerator at 3–5 °C for up to 20 days	The color and tenderness of beef samples as well as the water-holding capacity (WHC) percentage decreased with an increase in UV irradiation dose, while the 2-thiobarbituric acid (TBA) level increased.	[92]
Chicken meat	UV-C irradiation	0.62, 1.13, and 1.95 mW/cm²	Packaged in polyvinyl chloride film and kept at 4 °C for 9 days	The most stable b* values were 1.13 and 1.95 mW/cm^2^. L*, a*, pH, and TBARS values were also similar across groups.	[65]
*Tegillarca granosa* meat	Electron-beam irradiation	0, 1, 3, 5, 7 or 9 kGy	Vacuum-packed	The texture of *Tegillarca granosa* meat was increased by irradiation at 3–5 kGy.	[101]
*Nellore* beef	Gamma irradiation and aging combination	0, 3, 6, and 9 kGy	Vacuum-packed polyethylene	The combination of 9 kGy irradiation and aging improved tenderness.	[113]

**Table 4 foods-12-01845-t004:** Impact of irradiation on the shelf-life of meat.

Sample	Treatment	Dose	Sample Storage Conditions	Shelf-Life	Ref.
Beef, Chicken and Salmon Fillets	UV-C irradiation	360 J/m^2^	Vacuum storage; 5 days	A combination of vacuum packaging and UV irradiation extended the sample’s shelf life by 66.6%.	[54]
Broiler meat	Gamma irradiation	0, 2 and 3.5 kGy	Stored at −20 °C for 60 days	Broiler meat lasted longer after 2 kGy gamma irradiation.	[126]
Ground Beef Patties	Gamma irradiation	0.0, 1.0, 3.0, 5.0, and 7.0 kGy	Packed in container boxes; stored at 4 °C or −18 °C; maximum 42 days	Irradiation at 5 and 7 kGy increased the storage life of meat samples at 4 °C.	[132]
Chicken meat	E-beam irradiation	3, 4, 5, and 7 kGy	Vacuum-packed in LDPE bags	Irradiation at 4 kGy extended chicken meat shelf life without affecting sensory characteristics.	[131]
Marinated pork loin	E-beam irradiation	0.2–3 kGy	Stored in insulated boxes at 4–8 °C; 0->20 days	Samples stored at 4 °C lasted 7–16 days with a 1 kGy dose. A 2 kGy dose improved shelf life to 20 days.	[133]
Chicken meat	Gamma irradiation, 0.1 % chitosan and their combination	0.5 kGy	Packed in polypropylene bags, and stored in chilled conditions at 4–6 °C; storage for up to 14 days.	With 0.5 kGy gamma irradiation and 0.1% chitosan, chicken meat lasted 14 days in chilled storage.	[96]
Chicken breast meat	Gamma irradiation	4 kGy	Packaged under MAP; refrigeration storage at 2–6 °C; storage period of 25 days	Compared to air-packed samples, MAP (70% CO_2_/30% N_2_) and 4 kGy irradiation extended shelf life by 12 days.	[97]
Bovine meat	Gamma irradiation	0, 0.5, 1, 2 and 3 kGy	Commercial freezing bags; storage in the refrigerator at 4–7 °C 3 for weeks and in the freezer at −18 °C for 8 months	Irradiated samples had a shelf life of 14 days when stored at 4–7 °C and exposed to 3 kGy, while non-irradiated samples lasted only three days.	[45]
Meatballs	Gamma irradiation	20, 25 dan 35 kGy	Vacuum packed + frozen for 24 h; room temperature storage (irradiated samples) and freezer storage (non-irradiated); storage for 2 months	The quality of meatballs was maintained for two months at room temperature with 35 kGy irradiation.	[99]
Quail meat	Electron-beam irradiation	1.5, 3, and 5 kGy	Stored at 3–5 °C for 15 days	To maintain quail meat quality and shelf life, 1.5 and 3 kGy irradiation doses are recommended.	[75]

**Table 5 foods-12-01845-t005:** Impact of irradiation on nutritional characteristics of meat.

Sample	Treatment	Dose	Sample Storage Conditions	Nutritional Characteristics	Ref.
Goat meat	Gamma irradiation	0, 1, 2, 4 and 6 kGy	Packed in polystyrene foam boxes, at 2–4 °C	Irradiation increased the core nutritional content of DHA-phosphatidylcholine. The triacylglycerol levels significantly increased after irradiation, while the diacylglycerol levels decreased.	[154]
Chicken meat	E-beam irradiation	3, 4, 5, and 7 kGy	Vacuum-packed in LDPE bags	The vitamin C content decreased following e-beam irradiation at 4 kGy.	[131]
Chicken, turkey and mixed ground meat	X-ray irradiation	0.5, 1, 3, and 5 kGy	Plastic bag storage at −80 °C	Irradiation did not have any impact on the levels of free amino acids.	[125]
Chicken, turkey and mixed ground meat	X-ray irradiation	0.5, 1, 3, and 5 kGy	Plastic bag storage at −80 °C	Following irradiation, taurine became detectable, while the levels of glutathione decreased. Additionally, there was a formation of free fatty acids, but it was relatively insignificant.	[125]
Sliced ham	UV-C irradiation	408, 2040, 4080, and 6120 mJ/cm^2^	Refrigerated storage for 0, 7, and 14 days	The antioxidant capacity was higher on day 0 when irradiated with a 4080 mJ/cm^2^.	[52]
Spicy beef jerky	Gamma irradiation	0, 0.5, 1.5, 3, 4, 6, and 8 kGy	Vacuum-packed; refrigerator storage (4–8 °C) for 1 week	As the irradiation dose increases, free radicals were formed.	[62]
Spicy yak jerky	Electron-beam irradiation	0, 2, 5, 7 and 9 kGy	Vacuum-packed, and stored in the refrigerator at 4–8 °C for 1 week	The protein nutrition was significantly reduced at a high irradiation dose of 9 kGy, while there was no significant impact at doses ranging from 0–7 kGy.	[141]
Chicken Meat	Gamma irradiation	3 kGy	Aerobically packaged; refrigerator storage at 4 °C for up to 14 days	Irradiating chicken meat samples showed reduced amino and fatty acid loss during storage.	[94]
Smoked duck meat	Electron-beam irradiation	0, 1.5, 3.5, and 4.5 kGy	Vacuum-packed; stored at 4 °C for up to 40 days	The quality of smoked duck flesh, including its amino acids, fatty acids, and volatiles, did not change significantly during storage when exposed to <3 kGy irradiation.	[81]

**Table 6 foods-12-01845-t006:** Impact of irradiation on sensory characteristics of meat.

Sample	Treatment	Dose	Sample Storage Conditions	Sensory Characteristics	Ref.
Smoked guinea fowl meat	Gamma irradiation	2.5, 5 and 7.5 kGy	Stored in the refrigerator (2–4 °C) with a storage period of 7 weeks	Irradiation did not affect aroma, color, tenderness, or taste.	[51]
Sliced ham	UV-C irradiation	408, 2040, 4080, and 6120 mJ/cm^2^	Refrigerated storage; 0, 7, and 14 days	The 7th and 14th days of storage showed a slight color change, but consumers did not mind.	[52]
Fish and red meat-based ready-to-eat foods	Gamma irradiation	8 dan 45 kGy	Styrofoam boxes + ice block: non-irradiated foods. Plastic container: room temperature-irradiated foods. Menu cycle: 5 days for 30 days	The sensory characteristics of irradiated food, including overall appearance, texture, color, taste, and aroma, were deemed acceptable by consumers.	[161]
Marinated pork loin	E-beam irradiation	0.2–3 kGy	Stored in insulated boxes at 4–8 °C for 0->20 days	Prolonging the shelf life of irradiated products did not lead to any sensory quality alterations.	[133]
Korean native cattle (*Hanwoo*) beef	UV irradiation	4.5 mW s/cm^2^	Wrapped in polyvinyl chloride; stored at 3–5 °C for 9 days	The results of the sensory test showed no discernible difference between the irradiated and non-irradiated samples.	[95]
Rabbit meat	Gamma irradiation	0, 1.5, and 3 kGy	Packed in polyethylene pouches; stored in the refrigerator 3–5 °C for 3 weeks	Gamma irradiation did not significantly affect rabbit meat’s sensory properties.	[69]
Turkey breast meat	Gamma irradiation	0.0, 0.5, 2.0 and 4.0 kGy	Stored at −18 °C for 2 months	The preference scores given by the panelists for irradiated and non-irradiated samples were the same, indicating that both were equally acceptable in terms of appearance and smell.	[68]
Blue swimming crab meat (*Portunas pelagicus*)	Gamma irradiation	2, 4, and 6 kGy	Packed in polypropylene bags; stored at 0–4 for up to 28 days	Irradiation at 2–6 kGy in crab meat impacted the overall difference in odor quality.	[66]
Broiler meat	UV irradiation and activated oxygen (AO)	9.4, 18.8 and 32.9 mW/s per cm^2^	Vacuum bags; stored at 3–5 °C for up to 12 days	The sensory quality of samples not irradiated, UV-irradiated, or UV+AO-irradiated was not significantly different.	[61]
Raw ground beef	Electron-beam irradiation	0, 1.5, 3.0 and 4.5 kGy	Vacuum-packed and stored at 4 °C for 24 h	Exposing the sample to less than 3 kGy irradiation led to minor alterations in the sensory characteristics.	[56]
Ground beef	Gamma irradiation	First assay: 0.26, 0.44, 0.67 and 0.86. Second assay: 1 kGy. Third assay: 2.5 kGy.	Polyethylene bags; freezer storage at −18 °C	The irradiation of samples with 2.5 kGy gamma rays did not affect consumer acceptance results.	[57]
Minced camel meat	Gamma irradiation	0, 2, 4, and 6 kGy	Refrigerator storage at 1–4 °C; 2, 4, and 6 weeks of storage	Irradiated meat samples had similar sensory qualities to non-irradiated ones.	[58]
Raw ground beef patties	Electron-beam irradiation	2, 4, and 6 kGy	Packed with nylon polyethylene bag; stored at −18 °C; storage period of up to 7 months	The sensory characteristics of the beef samples were not affected by electron-beam irradiation.	[59]
Minced camel meat	Gamma irradiation	0, 2, 4 and 6 kGy	Placed in polystyrene trays and covered with polyethylene film; stored in refrigerator 1–4 °C; storage for up to 6 weeks	Irradiated camel meat had similar sensory properties to non-irradiated camel meat.	[58]
Beef trimmings (20% fat)	Gamma irradiation	2 to 5 kGy	Sterile bags (Whirl Pak^®^); polyethylene bags; stored at (− 18 ± 2) °C or (2 ± 2) °C for up to 30 days	No sensory differences were observed between the patties made from irradiated trimmings aged for 1 or 30 days.	[60]
Spicy beef jerky	Gamma irradiation	0, 0.5, 1.5, 3, 4, 6, and 8 kGy	Vacuum-packed; refrigerator storage (4–8 °C) for 1 week	As the irradiation dose increased, the preference for the color and taste of the spicy beef samples decreased.	[62]

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
