# Peer review of "Meat Irradiation: A Comprehensive Review of Its Impact on Food Quality and Safety"

_foods, 2023, doi:10.3390/foods12091845_

Round 1

Reviewer 1 Report

This was a vague review paper which has many shortcomings in terms of mechanical assembly of scientific opinion and literature evidence without any contribution to the scientific point of view in the field of meat irradiation. Similar review papers have been published comprehensively elsewhere.

Suggest to improve the English quality.

Author Response

Response to Reviewer 1 Comments

This was a vague review paper which has many shortcomings in terms of mechanical assembly of scientific opinion and literature evidence without any contribution to the scientific point of view in the field of meat irradiation. Similar review papers have been published comprehensively elsewhere.

Point 1: Suggest to improve the English quality

Response 1: Thanks for your suggestions. Improvements and more detailed discussion in each table have been carried out (revisions are marked in red on the manuscript). Improving the English quality has also been done by proofreading using Native, which has collaborated with my institution (Certificate attached).

Reviewer 2 Report

The review by Indriato et al comprehensively summarize the current available literature on meat irradiation. The language is clear and easy to understand. The authors have done a great job by referring more than 153 articles on the same topics.  

I have following suggestions for its further improvement as

        i.            L11: harmful microorganisms- it reduces microorganisms whether harmful of beneficial, so better if author delete the harmful word.

      ii.            L11-12: seems repetition

    iii.            L24: this product shoud be the products

    iv.            Keyword: as article has good discussion on met quality, so I recommned to add this in the keyword list.

      v.            L60 please mention wavelength of ionizing radiation

    vi.            Under chapter 2: please add information the measurement of radition (unit of radiation).

  vii.            L106: the scientific names of microbes in italics plz

viii.            Table 1, ref 49 row, aerobic bacteria, lactic acid bacteria-in non-italics; in same table ref 65 row, plz mention the unit for radiation, at 68 ref row, staph aur-in italics

    ix.            Section 4 and section 5: Seems too brief and need further detailing

English is good, only minor editing/ careful reading required

Author Response

Response to Reviewer 2 Comments

The review by Indiarto et al comprehensively summarize the current available literature on meat irradiation. The language is clear and easy to understand. The authors have done a great job by referring more than 153 articles on the same topics.  I have following suggestions for its further improvement as:

Point 1: L11: harmful microorganisms- it reduces microorganisms whether harmful of beneficial, so better if author delete the harmful word.

Response 1: Thanks for your suggestions. This section has been corrected by removing the word harm. Revisions are marked in red (line 11).

Point 2: L11-12: seems repetition

Response 2:  Thanks for your suggestions. The sentences in lines 11 and 12 have different meanings; the sentences in line 12 explain the sentences in line 11. The sentences have been revised to: This technology effectively reduces microorganisms such as viruses, bacteria, and parasites. It also increases the lifespan and quality of products by delaying spoilage and reducing the growth of microorganisms 

Point 3:  L24: this product shoud be the products

Response 3: Thanks for your suggestions. This section has been revised into a “products” (revision is marked in red, line 24), and the manuscript has been proofread.

Point 4: Keyword: as article has good discussion on met quality, so I recommned to add this in the keyword list.

Response 4: Thanks for your suggestions. In the keyword section, meat quality has been added. Revision is marked in red, line 27

Point 5: L60 please mention wavelength of ionizing radiation

Response 5: Thanks for your suggestions. The wavelengths have been mentioned in Figure 1(a), lines 100-101

Point 6: Under chapter 2: please add information the measurement of radition (unit of radiation).

Response 6: Thanks for your suggestions. Information about the unit of radiation has been added to the manuscript. Revisions are marked in red (lines 60-62).

Point 7: L106: the scientific names of microbes in italics plz

Response 7: Thanks for your suggestions. The scientific names for the microbes are italicized: Salmonella, Escherichia coli, Campylobacter, and Listeria monocytogenes (marked in red, line 107).

Point 8: Table 1, ref 49 row, aerobic bacteria, lactic acid bacteria-in non-italics; in same table ref 65 row, plz mention the unit for radiation, at 68 ref row, staph aur-in italics

Response 8: Thanks for your suggestions. In Table 1, ref. 49 for aerobic bacteria, lactic acid bacteria are not italicized (corrected, marked in red, ref. 54) and in ref. 65 for kGy dosage units (already added, marked in red, ref. 70), and in ref. 68 microbial names in italics for Escherichia coli or Staphylococcus aureus (corrected, marked in red, ref. 73)

Point 9: Section 4 and section 5: Seems too brief and need further detailing

Response 9: Thanks for your suggestions. More detailed information has been added to sections 4 and 5. Revisions are marked in red (lines: 419-430; lines: 446-459).

Comments on the Quality of English Language: English is good, only minor editing/ careful reading required.

Thanks for your suggestions. For the English language, the manuscript has been proofread by a native who has collaborated with my institution (Certificate attached).

Reviewer 3 Report

The current paper reviews te literature regarding the Meat Irradiation: A Comprehensive Review of Its Impact on Food Quality and Safety. 

The paper is nicely written, comprehensive and coherent. 

Although no major changes are needed, my only general suggestion for authors is to discuss a little more each table. Specifically to explore and explain the results from the papers they have reviewed. 

Which one of them obtained the best results and how, compared with others. 

Best of luck! 

Author Response

Response to Reviewer 3 Comments

The current paper reviews te literature regarding the Meat Irradiation: A Comprehensive Review of Its Impact on Food Quality and Safety.

The paper is nicely written, comprehensive and coherent.

Although no major changes are needed, my only general suggestion for authors is to discuss a little more each table. Specifically to explore and explain the results from the papers they have reviewed.

Which one of them obtained the best results and how, compared with others.

Best of luck!

Response 1: Thanks for your suggestions. A more detailed exploration and discussion of each Table have been added to the manuscript. Revisions are marked in red on the manuscript.

Round 2

Reviewer 1 Report

This work is of better quality than previous one whereby it deserves to be published on Foods journal.

Generally acceptable